# Effectiveness of EUS-Guided Fine-Needle Biopsy versus EUS-Guided Fine-Needle Aspiration: A Retrospective Analysis

**DOI:** 10.3390/diagnostics11060965

**Published:** 2021-05-27

**Authors:** Naosuke Kuraoka, Satoru Hashimoto, Shigeru Matsui, Shuji Terai

**Affiliations:** 1Department of Gastroenterology, Saiseikai Kawaguchi General Hospital, Kawaguchi 332-8558, Japan; shashim@saiseikai.gr.jp (S.H.); mshigerum@saiseikai.gr.jp (S.M.); 2Department of Gastroenterology and Hepatology, Graduate School of Medicine and Dental sciences, Niigata University, Niigata 951-8510, Japan; terais@med.niigata-u.ac.jp

**Keywords:** endoscopic ultrasound-guided fine-needle aspiration, endoscopic ultrasound-guided fine-needle biopsy, pancreatic tumor, subepithelial lesions

## Abstract

Endoscopic ultrasound-guided fine-needle aspiration (EUS-FNA) for pancreatic tumors and subepithelial lesions (SEL) of the gastrointestinal tract are effective for histological diagnosis. There are also reports that tissue sampling is possible with a smaller number of punctures by EUS-guided fine-needle biopsy (EUS-FNB). In this study, we retrospectively compared the diagnostic abilities of EUS-FNA and EUS-FNB. We examined 130 patients who underwent EUS-FNA/EUS-FNB for pancreatic tumors and SEL from July 2018 to January 2021. None of the cases underwent rapid on-site evaluation. There were 94 and 36 cases in the EUS-FNA and EUS-FNB groups, respectively. The median tumor size in the EUS-FNB group was 30 mm, which was significantly larger than the EUS-FNA group (*p* = 0.02). In addition, transgastric puncture was significantly more common in the EUS-FNB group (*p* = 0.01). The EUS-FNA and EUS-FNB groups had a sensitivity of 82.9% and 91.7% and an accuracy rate of 85.1% and 91.7%, respectively. However, both procedures had a comparable diagnostic ability.

## 1. Introduction

Endoscopic ultrasound-guided fine-needle aspiration (EUS-FNA) for pancreatic tumors and subepithelial lesions (SEL) of the gastrointestinal tract is effective for histological diagnosis [1,2,3,4]. In recent years, several puncture needles have been developed for increased tissue sampling. In particular, tissue sampling using a Franseen needle is called EUS-guided fine-needle biopsy (EUS-FNB), which is used to sample many tissues with a small number of punctures [5]. Accurate histological diagnosis may provide accurate treatment, and a large number of tissue samples may be used for immunostaining and gene mutation identification.

Rapid on-site evaluation (ROSE) can improve the accuracy rate of diagnosis by confirming the presence or absence of tissue samples during examination [6,7,8,9]. For accurate tissue diagnosis, the introduction of ROSE is desirable. However, at present, ROSE cannot be introduced at many facilities because of the long inspection times and a shortage of human resources. With EUS-FNB, sufficient tissue sampling can be expected without ROSE. In this study, we performed EUS-FNA/EUS-FNB without ROSE in pancreatic tumors and SELs to determine whether one procedure had better diagnostic abilities than the other, and we also examined the efficacy of different EUS-FNB needle types.

## 2. Materials and Methods

We retrospectively examined patients who underwent EUS-FNA or EUS-FNB for pancreatic tumors and SEL in our hospital from July 2018 to January 2021. EUS-FNA/EUS-FNB were performed for the purpose of histological diagnosis because of the limited number of cases in which pancreatic tumors and SELs were diagnosed by imaging and endoscopy. Inclusion criteria included (1) pancreatic tumor/SEL for which EUS-FNA/EUS-FNB could be performed, (2) patient consent and (3) cases in which ROSE was not performed.

EUS-FNA/EUS-FNB were performed by an endoscopist skilled in endoscopic ultrasound (Figure 1). During EUS-FNA, the lesion was visualized by endoscopic ultrasonography GF-UCT260 (Olympus Medical Japan, Tokyo, Japan) and was punctured with an end-cutting needle. Samples were collected using the slow-pull method or the standard method with 20 mL of air pressure. Either a 22G or 25G end-cutting needle was used. Additionally, an EZshot3 plus (Olympus Medical Japan, Tokyo, Japan) puncture needle was used (Figure 2). The puncture was performed until the tissue was sampled macroscopically.

During EUS-FNB, the lesion was also visualized by endoscopic ultrasonography GF-UCT260 (Olympus Medical Japan, Tokyo, Japan) but was punctured with a Franseen needle. Samples were collected using the slow-pull method without suction pressure. Either a 22G or 25G Franseen puncture needle was used. Additionally, we also compared the effectiveness of the Sono Tip Topgain^Ⓡ^ (MediGlobe, Rohrdorf, Germany) (Figure 3) and Acquire^TM^ (Boston Scientific Japan, Tokyo, Japan) (Figure 4) puncture needles. The puncture was performed until the tissue was sampled macroscopically.

Pancreatic tumor cells were detected by cytological or histological diagnosis using the specimens obtained by EUS-FNA/EUS-FNB and were defined as positive upon diagnosis. Gastrointestinal SEL was defined as positive when tumor cells could be detected by cytological or histological diagnosis using the samples obtained by EUS-FNA/EUS-FNB. In cases where histological diagnosis could not be obtained by EUS-FNA/EUS-FNB, surgery was performed based on clinical signs or diagnostic imaging, and a definitive diagnosis was obtained from the resected specimen. Inoperable cases were confirmed by diagnostic imaging and clinical signs 3 months later. The examination time was defined as the period from endoscope insertion to removal.

The results are presented as numerical values (%), and continuous variables are presented as median values (range). Two consecutive variables were tested using the Mann–Whitney U test. For non-contiguous variables, the Wilcoxon signed-rank test was used. Statistical significance was set at *p* ≤ 0.05. Using SPSS (IBM, Japan), the sensitivity was calculated as the ratio of test-positive to disease-positive, and the specificity was calculated as the negative ratio of disease negatives.

## 3. Results

From July 2018 to January 2021, 130 cases that met the inclusion criteria for pancreatic tumors and SELs were examined with EUS-FNA or EUS-FNB. There were 94 and 36 cases in the EUS-FNA and EUS-FNB groups, respectively.

### 3.1. Patient Characteristics

The median age of the EUS-FNA group was 71 years, which did not differ significantly from the median age of the EUS-FNB group (71.5 years) (*p* = 0.30). The proportion of men in the EUS-FNB group was low (*p* < 0.01). The median tumor size was 22 and 30 mm in the EUS-FNA and EUS-FNB groups, respectively, and was larger in the EUS-FNB group (*p* = 0.02). In both groups, pancreatic ductal cancer was the most common disease (EUS-FNA, 80.9%; EUS-FNB, 58.3%) in the pancreatic tumor group, and a gastrointestinal stromal tumor was the most common in both groups in the SEL group. (Table 1).

### 3.2. Outcomes

The median procedure time was 13 min in the EUS-FNA group and 16.5 min in the EUS-FNB group; thus, the procedure time was significantly shorter in the EUS-FNA group (*p* = 0.01). One adverse event (a tumor infection of the teratoma after puncture) was observed in the EUS-FNA group.

The median number of punctures was 2 in both groups; no significant difference was observed (*p* = 0.62). Transgastric puncture was the most common puncture site, with 62.8% in the EUS-FNA group and 77.8% in the EUS-FNB group. Thus, the EUS-FNB group had significantly more transgastric punctures than the EUS-FNA group (*p* = 0.01). In addition, puncture from the duodenum was significantly higher in the EUS-FNA group in both the bulb and descending parts (bulb, *p* < 0.01; descending, *p* < 0.01). The 22G puncture needle was used in 91.1% of cases in the EUS-FNA group and 100% of cases in the EUS-FNB group (Table 2).

### 3.3. Subgroup Analysis of SEL

Regarding the patient backgrounds of SEL, there was no significant difference between the EUS-FNA group and the EUS-FNB group in terms of male proportion and age. There was no difference in tumor size in the SEL subgroup between the two groups. (*p* = 0.28) There was no significant difference between the EUS-FNA group and the EUS-FNB group at the puncture site. The number of punctures was significantly smaller in the EUS-FNB group. (*p* = 0.04) (Table 3).

### 3.4. Sensitivity, Specificity and Accuracy of EUS-FNA and EUS-FNB

The overall sensitivity was 82.9% (68/82) in the EUS-FNA group and 91.4% (32/35) in the EUS-FNB group. No false negatives were found, and the specificity was 100% in both groups. The accuracy rate was 85.1% (80/94) in the EUS-FNA group and 91.7% (33/36) in the EUS-FNB group. The accurate diagnosis rate of pancreatic tumors in the EUS-FNA and EUS-FNB groups was 81.6% (62/76) and 85.7% (18/21), respectively. The accurate diagnosis rate of SEL lesions was 100% in both groups (Table 4).

### 3.5. EUS-FNB Needle Types

The diagnostic ability of the two different types of EUS-FNB needles was also examined. The sensitivity of Acquire^TM^ and Sono Tip Topgain^Ⓡ^ were 91.5% (22/24) and 90.9% (10/11), respectively. The accuracy rates for Acquire^TM^ and Sono Tip Topgain^Ⓡ^ were 88.0% (22/25) and 90.9% (10/11), respectively (Table 5).

## 4. Discussion

In this study, we examined the patient characteristics and diagnostic abilities of EUS-FNA and EUS-FNB. The tumor size was larger in the EUS-FNB group than the EUS-FNA group, and the EUS-FNB group had more transgastric punctures than the EUS-FNA group in the whole group. Franseen-shaped needles used in EUS-FNB had inferior penetrability to end-cutting needles and were used for transgastric punctures and large tumor-sized lesions due to the poor puncture properties; it was thought that momentum was required during the puncture. In contrast, puncture from the duodenum was significantly more frequent in the EUS-FNA group than the EUS-FNB group, and puncture with an end-cutting needle was selected to avoid organs such as the pancreatic duct and blood vessels around the duodenum.

On the other hand, interesting results were obtained when the analysis was performed only on the SEL group. In SEL, there was no significant difference in tumor size between the EUS-FNA group and the EUS-FNB group. No significant difference was observed in the puncture site between the two groups, and the puncture from the stomach was slightly more frequent in the EUS-FNB group. The number of punctures was significantly smaller in the EUS-FNB group. From the above, it was shown that in both the EUS-FNA group and the EUS-FNB group, SEL was punctured regardless of the tumor size and tumor site, but the number of punctures was smaller in the EUS-FNB group. The reason is considered to mean that more samples are required for immunostaining in SEL lesions, and more samples were obtained in the EUS-FNB group.

Both procedures had comparable results; however, the accurate diagnosis rate was higher in the EUS-FNB group. Furthermore, a comparison of the EUS-FNB needle types revealed that the Acquire^TM^ needle (a Franseen needle that has been used for a long time) had higher sensitivity, but the Sono Tip Topgain^Ⓡ^ needle (the latest Franseen needle developed in 2020) had a higher accuracy rate.

Tissue sampling using EUS-FNA or EUS-FNB has been widely documented. Many meta-analyses reveal that, in terms of the diagnostic results, EUS-FNA on pancreatic tumors has a sensitivity of approximately 90% and a specificity of 95% or more [1,2,3,4]. However, some comparisons between EUS-FNA and EUS-FNB reveal that there are no differences in diagnostic ability between the two procedures [10,11,12,13]. On the other hand, there are few reports on the comparison between EUS-FNA and EUS-FNB for SEL. According to reports, the diagnostic ability of EUS-FNB is high, but the number of cases is still small [14,15].

There are also many reports on ROSE results for solid pancreatic masses. While some reports reveal that ROSE improves diagnostic ability, others reveal that it does not contribute to the accurate diagnosis rate; thus, whether ROSE improves diagnostic ability remains controversial [6,7,8,9].

In this study, EUS-FNA/EUS-FNB were performed without ROSE, and the results were comparable to those reported previously. Since both groups produced the same number of punctures, the introduction of ROSE is not essential.

Recently, several puncture needles for EUS-FNB have been developed for endoscopic visibility and/or gastrointestinal puncture ability. Many reports on the diagnostic ability of EUS-FNB in recent years have revealed that the accurate diagnosis rate is high, between 91.6% and 98.7%, and that this procedure provides accurate diagnosis for large tumor sizes. It has also been reported that the number of punctures in EUS-FNB is less than EUS-FNA [16,17,18]. However, we did not observe differences in diagnostic performances between the procedures, and the different EUS-FNB needles also had a comparable diagnostic performance. Besides, this study showed that EUS-FNB could be diagnosed with a small number of punctures for SEL.

EUS-FNB may be often applied to large lesions that are easy to puncture in the whole disease, suggesting that it has poor penetration compared to end-cutting needles. Thus, EUS-FNA should be selected when important organs are in close proximity. Although adverse events related to EUS-FNA are as low as 1.7%, and no fatal adverse events have been reported, it is necessary to be familiar with the characteristics of the puncture needle to avoid adverse events and to sample many tissues [19].

This study is not without limits. First, this was a retrospective single-center report. Future studies could investigate whether the diagnostic abilities of EUS-FNA and EUS-FNB differ on a multi-center level. Second, there were fewer cases for EUS-FNB than EUS-FNA. Therefore, future research with larger case numbers is necessary.

## 5. Conclusions

EUS-FNA and EUS-FNB without ROSE had equivalent diagnostic abilities. In SEL, the number of punctures may be smaller in the EUS-FNB group than in the EUS-FNA group. Nonetheless, there was no difference in diagnostic performance between the different EUS-FNB needle types.

## Figures and Tables

**Figure 1 diagnostics-11-00965-f001:**
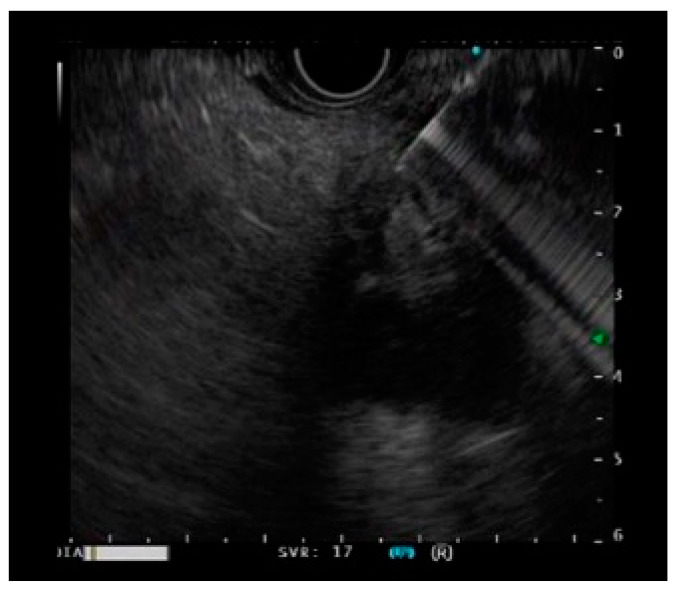
A case of a pancreatic tumor undergoing EUS-guided fine-needle biopsy.

**Figure 2 diagnostics-11-00965-f002:**
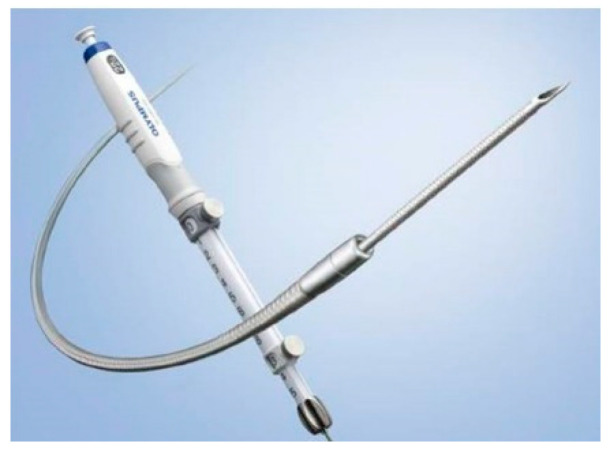
The EZshot3 plus (Olympus Medical Japan, Tokyo, Japan).

**Figure 3 diagnostics-11-00965-f003:**
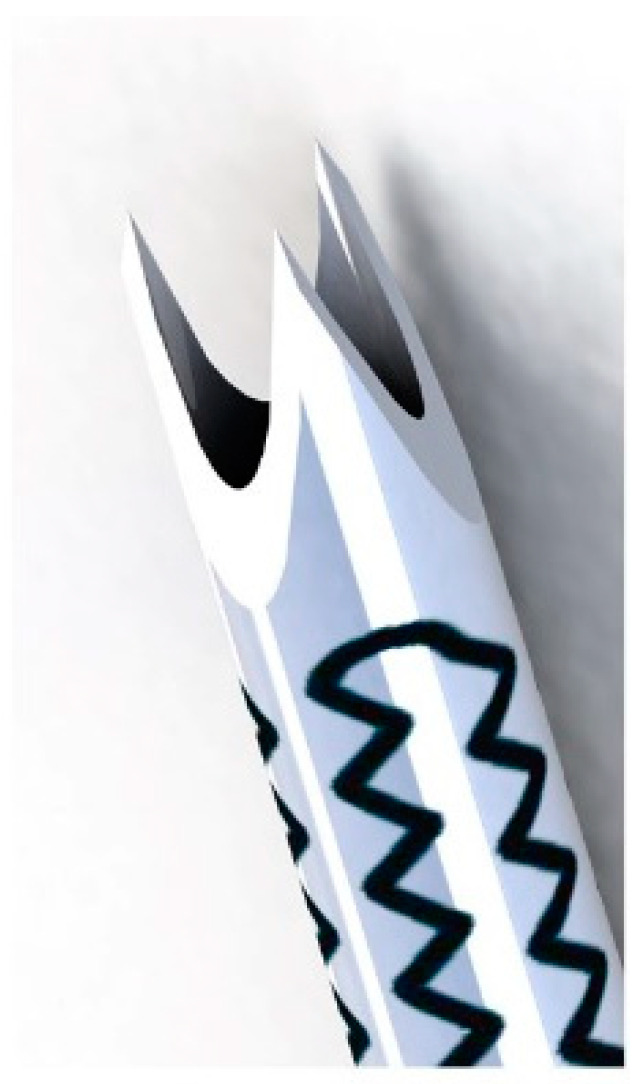
The Sono Tip Topgain^Ⓡ^ (MediGlobe, Rohrdorf, Germany).

**Figure 4 diagnostics-11-00965-f004:**
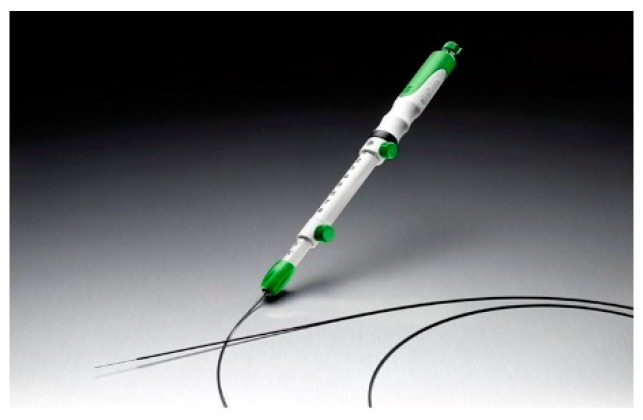
The Acquire^TM^ (Boston Scientific Japan, Tokyo, Japan).

**Table 1 diagnostics-11-00965-t001:** Patient characteristics of EUS-FNA and EUS-FNB groups.

	EUS-FNA (*n* = 94)	EUS-FNB (*n* = 36)	*p*-Value
Sex, Male(%)	56 (59.6)	14 (38.9)	<0.01
Age, median (range)	71 (38–86)	71.5 (43–90)	0.3
Tumor size, mm, median (range)	22 (6–70)	30 (12–70)	0.02
diagnosis			
Pancreas tumors, *n*(%)	76 (80.9)	21 (58.3)	0.05
Pancreatic ductal cancer, *n*(%)	54 (57.4)	16(44.4)	
Metastatic pancreatic tumor, *n*(%)	4 (4.3)	1 (2.8)	
Pancreatic neuro-endocrine tumor, *n*(%)	6 (6.4)	1 (2.8)	
Acinar cell carcnioma, *n*(%)	0	1 (2.8)	
Tumor-foming pancreatitis, *n*(%)	12 (12.8)	1 (2.8)	
B cell lymphoma, *n*(%)	0	1 (2.8)	
Subepithelial lesions, *n*(%)	18 (19.1)	15 (41.6)	0.11
Gastrointesitnal stromal tumor, *n*(%)	9 (9.6)	13 (36.1)	
Leiomyoma, *n*(%)	4 (4.3)	0	
Schwanomma, *n*(%)	4 (4.3)	2 (5.6)	
Teratoma, *n*(%)	1 (1.1)	0	

**Table 2 diagnostics-11-00965-t002:** Outcomes of EUS-FNA and EUS-FNB.

	EUS-FNA (*n* = 94)	EUS-FNB (*n* = 36)	*p*-Value
Procedure time, min, median (range)	13 (5–36)	16.5 (6–40)	0.01
Adverse events, *n*(%)	1 (1.1)	0 (0)	
Number of punctures, median(range)	2 (1–3)	2 (1–3)	0.62
Puncture site			
Esophagus, *n*(%)	2 (2.1)	0	
Rectum, *n*(%)	1 (1.1)	0	
Stomach, *n*(%)	59 (62.8)	28 (77.8)	0.01
Duodenal bulb, *n*(%)	17 (18.1)	6 (16.7)	<0.01
Descending part of duodenum, *n*(%)	15 (16)	2 (5.6)	<0.01
Needle			
EZ shot3 plus			
22G, *n*(%)	86 (91.1)	N/A	
25G, *n*(%)	8 (8.9)	N/A	
Acquire^TM^, 22G, *n*(%)	N/A	25 (73.3)	
Topgain^Ⓡ^, 22G, *n*(%)	N/A	11 (26.7)	

**Table 3 diagnostics-11-00965-t003:** Subgroup analysis of SELs.

	EUS-FNA (*n* = 18)	EUS-FNB (*n* = 15)	*p*-Value
Sex, Male(%)	10 (55.6)	8 (53.3)	0.18
Age, median (range)	68 (42–82)	68 (43–80)	0.87
Tumor size, mm, median (range)	28.5 (17–70)	20 (6–40)	0.28
Puncture site			
Esophagus, *n*(%)	2 (11.1)	0	
Rectum, *n*(%)	1 (5.6)	0	
Stomach, *n*(%)	12 (66.7)	13 (86.7)	0.31
Duodenal bulb, *n*(%)	2 (11.1)	2 (13.3)	
Descending part of duodenum, *n*(%)	1 (5.6)	0	
Number of punctures, median(range)	2.5 (1–3)	2 (1–2)	0.04

**Table 4 diagnostics-11-00965-t004:** The sensitivity, specificity and accuracy of EUS-FNA and EUS-FNB.

	EUS-FNA (*n* = 94)	EUS-FNB (*n* = 36)
Overall sensitivity, %	82.9 (68/82)	91.4 (32/35)
Overall specificity, %	100 (12/12)	100 (1/1)
Overall accuracy, %	85.1 (80/94)	91.7 (33/36)
Pancreatic tumors		
Sensitivity, %	78.1 (50/64)	85 (17/20)
Specificity, %	100 (12/12)	100 (1/1)
Accuracy, %	81.6 (62/76)	85.7 (18/21)
Subepithelial lesions		
Sensitivity, %	100 (18/18)	100 (15/15)
Specificity, %	N/A	N/A
Accuracy, %	100 (18/18)	100 (15/15)

**Table 5 diagnostics-11-00965-t005:** The sensitivity, specificity and accuracy of EUS-FNB group.

EUS-FNB (*n* = 36)	Acquire^TM^ (*n* = 25)	Sono Tip Topgain^Ⓡ^ (*n* = 11)
Overall sensitivity, %	91.6 (22/24)	90.9 (10/11)
Overall specificity, %	100 (2/2)	N/A
Overall accuracy, %	88 (22/25)	90.9 (10/11)

## Data Availability

The data presented in this study are available on request from the corresponding author. The data are not publicly available due to ethical and privacy restrictions.

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
