# Peer review of "Effectiveness of EUS-Guided Fine-Needle Biopsy versus EUS-Guided Fine-Needle Aspiration: A Retrospective Analysis"

_diagnostics, 2021, doi:10.3390/diagnostics11060965_

Round 1

Reviewer 1 Report

 Please describe: slow-pull method and 20 ml of air 53 pressure.

Please describe more exact risks and chances of the procedure

Author Response

Reviewer 1

   Thank you for a chance to revise our paper.

  •  Please describe: slow-pull method and 20 ml of air 53 pressure.

Thank you for great comment for our paper. It was a misnomer as slow-pull method or standard suction. We corrected it.

  • Please describe more exact risks and chances of the procedure.

Thank you for great comment for our paper. We added about the risks of adverse events in the manuscript.

Reviewer 2 Report

  • Since this is a retrospective non-randomized review, what is the determinant for choosing the type of needles about FNA or FNB (and also the determinant for which type of Franseen needle?)
  • Since ROSE was not performed , how did the endosonographer in this study determine the number of needle puncture? (decision of the adequacy of tissue specimen obtained)
  • It seems not the usual way of description for the inclusion criteria: “(3) cases of tumor puncture, and (4) cases in which ROSE was not performed”.
  • P1 L32: “EUS-FNB needles are considered to be useful despite as many reports describe the use of puncture needles with special tip shapes.” àThe sentence is confusing.
  • P5 L131~132:”The number of punctures was significantly smaller in the EUS-FNB group (p = 0.04)” (Table 3).--> However, the median of both EUS-FNA and EUS-FNB group were 2(1-2) and 2(1-3)
  • In table 1, there were 9 cases of GIST, 4 cases of leiomyoma and 1 case of teratoma diagnosed by EUS-FNA, what were the pathological diagnostic criteria for them by FNA needles?
  • In Table 2 the (%) for SEL is incorrect
  • In table 4: “pancres tumor”à should be “pancreatic tumor”

Author Response

Reviewer 2

   Thank you for a chance to revise our paper.

  • Since this is a retrospective non-randomized review, what is the determinant for choosing the type of needles about FNA or FNB (and also the determinant for which type of Franseen needle?)

Thank you for great comment for our paper, FNA tended to be selected when important organs such as blood vessels and bile ducts were nearby, but FNA and FNB were selected at random to some extent. The same choice is made for the Franseen needle.

  • Since ROSE was not performed, how did the endosonographer in this study determine the number of needle puncture? (decision of the adequacy of tissue specimen obtained)

Thank you for great comment for our paper. After puncturing, the number of punctures was determined macroscopically based on the presence or absence of a tissue. We described in the text.

  • It seems not the usual way of description for the inclusion criteria: “(3) cases of tumor puncture, and (4) cases in which ROSE was not performed”.

Thank you for great comment for our paper. We deleted the criteria in (3).

  • P1 L32: “EUS-FNB needles are considered to be useful despite as many reports describe the use of puncture needles with special tip shapes.” The sentence is confusing.

Thank you for great comment for our paper. We deleted this sentence from the manuscript.

  • P5 L131~132:”The number of punctures was significantly smaller in the EUS-FNB group (p = 0.04)” (Table 3).--> However, the median of both EUS-FNA and EUS-FNB group were 2(1-2) and 2(1-3)

Thank you for great comment for our paper. The median number of FNA puncture was calculated to be 2.5, and the median number ‘2’ was a clerical error. We fixed it.

  • In table 1, there were 9 cases of GIST, 4 cases of leiomyoma and 1 case of teratoma diagnosed by EUS-FNA, what were the pathological diagnostic criteria for them by FNA needles?

Thank you for great comment for our paper. We performed differential diagnosis using immunostaining. If KIT was positive, GIST was diagnosed, and if KIT was negative and desmin was positive, leiomyoma was diagnosed. Regarding teratoma, soft tissue was seen by HE staining and it was possible to diagnose.

  • In Table 2 the (%) for SEL is incorrect

Thank you for great comment for our paper. We corrected it.

  • In table 4: “pancres tumor”à should be “pancreatic tumor”

Thank you for great comment for our paper. We corrected it.